# Mutation Induction in Humans and Mice: Where Are We Now?

**DOI:** 10.3390/cancers11111708

**Published:** 2019-11-01

**Authors:** Yuri Dubrova

**Affiliations:** Department of Genetics and Genome Biology, University of Leicester, Leicester LE1 7RH, UK; yed2@le.ac.uk; Tel.: +44-116-252-5654

**Keywords:** mutation, germline, ionizing radiation, mutagens

## Abstract

The analysis of mutation induction in human families exposed to mutagens provides the only source of reliable estimates of factors contributing to the genetic risk of human exposure to mutagens. In this paper, I briefly summarize the results of recent studies on the pattern of mutation induction in the human and mouse germline. The results of recent studies on the genome-wide effects of exposure to mutagens on mutation induction in the mammalian germline are presented and discussed. Lastly, this review also addresses the issue of transgenerational effects of parental exposure to mutagens on mutation rates in their non-exposed offspring, which are known as transgenerational instability. The possible contribution of transgenerational instability to the genetic risk of human exposure to mutagens is discussed.

## 1. Introduction

Given that mutations induced in the germline can persist for a number of generations following parental exposure to mutagens, they may substantially contribute to the burden of inherited diseases and thus represent one of the key risk factors of human exposure to mutagens [1,2,3]. It should be stressed that to date some fundamental gaps remain in our understanding of the pattern of mutation induction in humans. The aim of this Review is, therefore, to present and discuss the progress made in the field of mutation induction in the mammalian germline and to examine its potential contribution to our current understanding of the factors contributing to the genetic risk of human exposure to mutagens.

## 2. Mutation Induction and Transgenerational Effects

### 2.1. Mutation Induction in the Human and Mouse Germline

Traditionally, the effects of exposure to ionizing radiation and mutagenic anticancer drugs was studied by analyzing the frequency of malformations and incidence of infant mortality among the children of exposed parents [1,2,3]. For example, the data collected so far on families of atomic bomb survivors [4] and cancer patients treated by radiotherapy or chemotherapy [5] did not provide any evidence for significant increases in the incidence of adverse effects among their children. The absence of significant effects may partially be attributed to the very low contribution of de novo mutations to infant mortality and morbidity [3]. As mentioned by the United Nations Scientific Committee on the Effects of Atomic Radiation, the results of these studies are at odds with those obtained by analyzing radiation-induced mutations in the germline of other species [1]. It is, therefore, clear that more sensitive approaches for monitoring germline mutation induction in humans should be developed.

The sensitivity of any approach for monitoring mutation induction depends on the rate of spontaneous mutation. If the number of de novo spontaneous mutations per offspring is high, then the effects of exposure to mutagens can be robustly established within a broad range of doses. In this respect, tandem repeat human mini satellite loci represent an attractive biomarker of exposure to mutagens as the spontaneous mutation rate at these loci by several orders of magnitude exceeds those for protein-coding genes [6,7]. In our studies, the mini satellite mutation rate was established in the germline of human families exposed to ionizing radiation following the Chernobyl accident in Belarus and Ukraine [8,9,10], to nuclear weapon tests in the vicinity of the Semipalatinsk nuclear weapon test site in Kazakhstan [11], and to multiple discharges of radioactive waste into the Techa River [12]. For all irradiated families, we detected a significant increase in the mini satellite mutation rate (Figure 1). The mini satellite mutation rate was also analyzed in the germline of irradiated families from Hiroshima and Nagasaki [13], Chernobyl clean-up workers [14,15], cancer survivors [16,17], and occupationally exposed workers at the Sellafield nuclear facility [18]. The results of these studies were negative and did not show that exposure to ionizing radiation significantly increased the mini satellite mutation rate in these families. As previously suggested [19], the negative results obtained by profiling the families of Chernobyl clean-up workers could be explained by the relatively low doses of fractionated exposure to ionizing radiation (0.25 Gy and below), the effects of which, on mutation induction, in the germline may be too small to be realistically detected.

Another group of highly unstable tandem repeat DNA loci (expanded simple tandem repeat loci, ESTRs) was found in the mouse genome [20]. The rate of spontaneous mutation at these loci is close to that for human mini satellites. In our studies, we have used ESTR loci for monitoring mutation induction in the germline of male mice exposed to ionizing radiation, chemical mutagens, anticancer drugs, and extremely low frequency magnetic fields [21,22,23,24,25,26,27,28,29]. The results of our studies highlight the key advantage of this technique that allows establishing the effects of exposure to mutagens within a very broad range of doses. For example, in the previous studies, mice were exposed to very high doses of anticancer drugs, often substantially above the clinically relevant doses for humans [30]. In contrast, in our study, for the first time, we analyzed mutation induction in the germline of male mice within the range of the clinically relevant doses [28], which provides important evidence for the risk of exposure to these drugs in humans. Table 1 summarizes the results of our mouse studies.

ESTR mutation induction was also analyzed in the germline of male mice exposed to a variety of mutagens, including tobacco smoke [31] and particulate air pollution [32,33,34]. In our studies, on knock-out mice, we established the combined effects of DNA-repair deficiencies and exposure to mutagens on the ESTR mutation rate in the mouse germline [35,36,37,38].

In summary, the results of our studies have shown that tandem repeat DNA loci provide a useful and convenient tool for monitoring radiation-induced mutations in humans. It should, however, be stressed that since the germline mutation at human mini satellites is almost exclusively attributed to complex meiotic gene-conversion-like events [40], the mechanism of mutation induction at these loci differs from that at protein-coding genes. It is well established that mutation induction at protein-coding genes mostly reflects direct targeting of these loci by ionizing radiation [41], whereas, at mini satellites, it results from the accumulation of DNA damage elsewhere in the genome, which, subsequently, affects the stability of mini-satellite loci at meiosis [8]. Given the profound differences in the mechanisms of mutation induction, the results of mini-satellite studies cannot, therefore, provide reliable estimates of the genetic risk of human exposure to ionizing radiation. For that reason, new systems for monitoring are needed to evaluate in greater detail radiation-induced mutation in the human germline.

Recent advances in whole-genome sequencing provided new tools for the analysis of germline mutation. In our recent study, we used comparative genome hybridization and whole-genome sequencing to establish the genome-wide pattern of mutation induction in the germline of male mice exposed to 3 Gy of acute X-rays [42]. The results of our study are summarized in Figure 2. We found that exposure to ionizing radiation significantly increases the rate of a spectacular variety of mutations, including large copy number variants (CNVs) and small insertion/deletion events. An interesting set of data was obtained by analyzing the rate of single nucleotide variants (SNVs) in the germline of irradiated males. It turned out that the frequency of de novo SNVs among the offspring of irradiated males was close to that in controls. On the other hand, we found a highly elevated frequency of clustered mutations containing several de novo mutation within a few base pairs of each other. Their elevated frequency among the offspring of irradiated males can be attributed to the delayed or compromised repair of radiation-induced clustered damaged sites [43]. The results of a recent study also show significant genome-wide changes in the spectrum of mutations in the germline of male mice exposed to chemical mutagen benzo(a)pyrene [44].

In summary, the results of genome-wide analysis of mutation induction in the mouse germline highlight the key advantages of this technique. First, since the total number of de novo mutations detected by the genome-wide analysis is quite high (>60 per offspring in humans [45,46]), next-generation sequencing has ample statistical power to detect mutation induction in small population samples and at relatively low doses of exposure. Most importantly, the genome-wide analysis of germline mutation allows characterizing, in sufficient detail, the spectra of spontaneous and induced mutations, which would provide important information regarding the potential impact of parental exposure on the health status of their children.

### 2.2. Transgenerational Effects of Parental Exposure to Mutagens

As already mentioned, mutation induction in the parental germline is regarded as the main component of the genetic risk of human exposure to mutagens, including ionizing radiation [1]. However, according to the results of some recent laboratory studies, parental exposure to mutagens may also destabilize the genomes of their non-exposed offspring. Since transgenerational instability affects the non-exposed offspring, it was recently ascribed to non-targeted effects of ionizing radiation [47].

One of the first experimental results for transgenerational effects was obtained by Luning and co-workers by analyzing the incidence of dominant mutations among the second-generation (F_2_) offspring of irradiated male mice [48]. The authors established the rate of dominant lethal mutations in the germline of non-exposed first-generation offspring (F_1_) of male mice exposed to the alpha-particle emitter plutonium-239. According to the results of this study, the F_1_ rate of dominant lethal mutations was highly significantly elevated. In 2000, the issue of transgenerational instability was revisited [49]. In this study, we established mutation rates at ESTR loci in the germline of first-generation offspring of male mice exposed to 0.5 Gy of fission neutrons and showed that parental irradiation destabilized the F_1_ genomes. These results were later confirmed by analyzing ESTR mutation rates in the germline of non-exposed F_1_ and F_2_ offspring of irradiated male mice from three different inbred strains [50].

In the previously mentioned studies, the mutation rate was measured in the germline. In our later studies, we also established ESTR mutation frequencies in the germline (sperm) and somatic tissues of F_1_ offspring of irradiated male mice [24,51]. Figure 3a summarizes our results, which shows that the manifestation of transgenerational instability is not tissue-specific. Of particular interest are the results of two studies showing that, in the offspring conceived by irradiated males and non-exposed females, the frequency of somatic mutations is equally elevated in both alleles derived from both the irradiated fathers and the unexposed mothers [51,52]. This points toward the genome-wide transgenerational destabilization.

We have also analyzed the transgenerational effects of maternal irradiation among the F_1_ offspring of irradiated female mice mated with non-exposed males [24,25]. In sharp contrast to the previously mentioned results obtained in the offspring of irradiated male mice, maternal irradiation did not affect the F_1_ genome stability. These results remain unexplained and future work should address the mechanisms underlying the differential transgenerational responses following paternal and maternal irradiation.

The results obtained by analyzing mutation rates in the offspring of irradiated males raised the important issue of transgenerational effects of parental exposure to chemical mutagens, which was addressed in our two studies on the effects of paternal exposure to the alkylating agent ethylnitrosourea (ENU) or three commonly-used anticancer drugs—cyclophosphamide, mitomycin C, and procarbazine [53,54]. The aim of our work was to establish whether an instability signal is triggered by a specific subset of DNA lesions such as by radiation-induced double strand breaks, or it can be attributed to generalized DNA damage following exposure to a wide range of mutagens. According to our data, the manifestation of transgenerational instability among the F_1_ offspring of male mice exposed to the previously mentioned anticancer drugs was close to that in the offspring of irradiated males (Figure 3a).

Judging from the results obtained by analyzing the offspring of exposed parents, the phenomenon of transgenerational instability is attributed to the yet unknown epigenetic mechanisms. Given that an equally elevated ESTR mutation rate was detected in practically all F_1_ offspring of irradiated males, it cannot, therefore, be explained by Mendelian segregation of de novo mutations induced in the germline of exposed parents [50]. Despite the fact that the mechanisms underlying the induction of an epigenetic transgenerational signal in the germline of exposed males and its transmission to the offspring remains unknown, the results of some recent publications have provided some insights regarding the factors affecting the F_1_ genome stability. In our study, the amount of endogenous DNA damage (single-stranded and double-stranded DNA breaks) was measured in the F_1_ offspring of control and irradiated male mice [51]. According to our data, the amount of endogenous DNA damage in the F_1_ offspring of irradiated males significantly exceeded that in controls. The intriguing observation for this study is that the efficiency of DNA repair in the offspring of irradiated males is not compromised. Considering these data, it would appear that destabilization of the F_1_ genome may somehow be related to deregulated DNA replication or a compromised apoptosis/cell cycle arrest. A detailed analysis of the expression profiles in F_1_ tissues should elucidate the still unknown mechanisms underlying the phenomenon of radiation-induced genomic instability.

Given that transgenerational destabilization of F_1_ genomes is a genome-wide phenomenon, a related question is whether it can affect a number of health-related traits in the offspring. The results of some publications suggest that it may be the case. As already mentioned, the data obtained by Luning and co-authors show that paternal irradiation can substantially increase the rate of dominant lethal mutations in the germline of non-exposed F_1_ offspring [48]. Increased incidence of cancer was found among carcinogen-challenged offspring of irradiated male rats and mice [55,56], which can be attributed to the transgenerational destabilization promoting tumor progression.

The results of previously mentioned animal studies may imply that transgenerational destabilization of F_1_ and, possibly, F_2_ genomes may be regarded as an additional component of the genetic risk of human exposure to mutagens. However, the manifestation of transgenerational effects among the children of irradiated parents has been analyzed only in a handful of studies [57,58]. Their results showing either an elevated [57] or a baseline [58] frequency of chromosome aberrations among the children of irradiated parents have not, therefore, provided coherent experimental evidence for the manifestation of transgenerational instability in humans. It should be noted that, in the previously mentioned animal studies, male mice were acutely exposed to high doses of ionizing radiation, which often is not the case for humans. In our study, we established the effects of low-doses/medium-doses of acute and chronic paternal exposure to γ-rays on the manifestation of transgenerational instability among their offspring [39]. According to our results, acute exposure to 10 cGy of γ-rays, the maximum dose to normal tissues per single radiotherapy procedure, does not affect the offspring (Figure 3b). On the other hand, paternal exposure to clinically-relevant doses of anticancer drugs can significantly destabilize the F1 genomes (Figure 3a). If correct, our data imply that, in most cases, the doses of parental exposure to ionizing radiation in humans are not high enough to destabilize the F_1_ genomes. It would also appear that high-dose parental exposure to some anticancer drugs could result in transgenerational effects among their children.

## 3. Conclusions

As already mentioned, recent advances in molecular genetics have provided highly sensitive tools for the genome-wide analysis of genetic variation. The whole-genome sequencing will provide fundamental quantitative information on the genetic effects of exposure to mutagens in humans and will, therefore, be important in improving the accuracy of the estimates of genetic risks of exposure to ionizing radiation and other mutagens in humans. It should be stressed that this technique will quantify the effects of parental exposure to mutagens and characterize the spectra of induced mutations found in the offspring of control and exposed parents. Considering the result of animal studies on transgenerational instability, further work aimed at establishing the mechanisms underlying this epigenetic phenomenon as well as its manifestation in humans are clearly warranted.

## Figures and Tables

**Figure 1 cancers-11-01708-f001:**
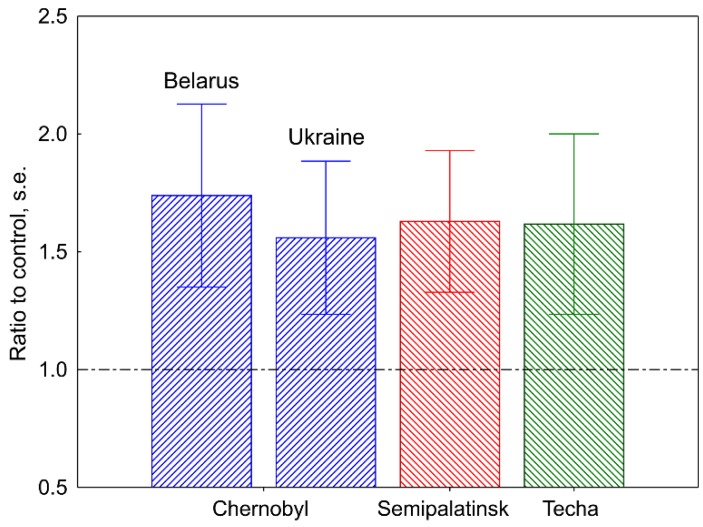
The mini satellite mutation rate in the germline of human families accidentally exposed to ionizing radiation. Data from References [8,9,10,11,12].

**Figure 2 cancers-11-01708-f002:**
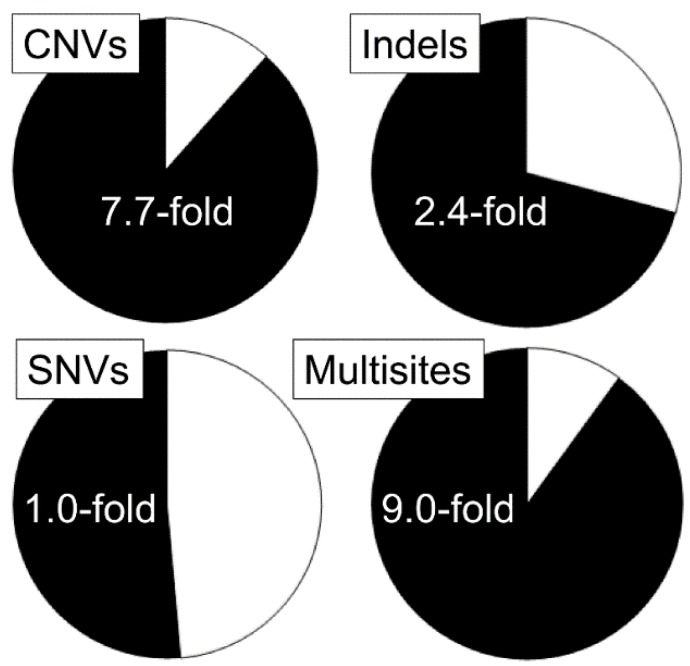
De novo mutation frequencies in the mouse germline established by the whole-genome comparative hybridization and next-generation sequencing. The frequency of de novo mutations per offspring of control (white) and irradiated males (black) are shown. Data from Reference [42].

**Figure 3 cancers-11-01708-f003:**
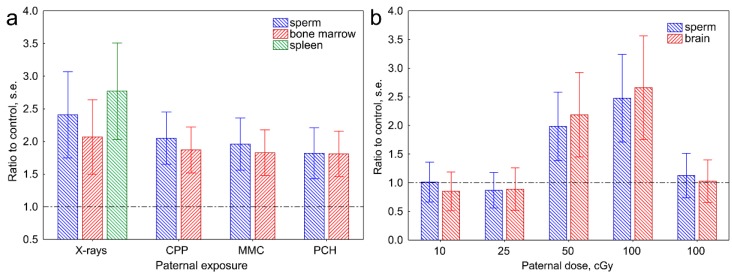
The transgenerational effects of paternal exposure to mutagens on ESTR mutation frequencies in the F_1_ offspring of exposed male mice. (**a**) Mutation frequencies in the F_1_ offspring of male mice exposed to acute X-rays, and anticancer drugs cyclophosphamide (CPP), mitomycin c (MMC), and procarbazine (PCH). Data from Reference [54]. (**b**) Dose and dose-rate effects on the manifestation of transgenerational instability in the F_1_ offspring of irradiated male mice. Data from Reference [39]. On all graphs, the ratio of ESTR mutation frequencies in the offspring of exposed males to that in control animals is shown.

**Table 1 cancers-11-01708-t001:** ESTR mutation induction in the germline of male mice.

Mutagen, Reference	Strains	Doses	Results ^1^
**Ionizing radiation:**			
Acute X-rays [22]	CBA/H	0.5–1 Gy	+
Acute γ-rays [39]	BALB/c	0.1–1 Gy	+
Chronic γ-rays [23,39]	CBA/H	0.5–1 Gy	+
Fission neutrons [23]	CBA/H		+
**Anti-cancer drugs:**			
Cisplatin [27]	F_1_(C57BL/6JxCBA/Ca)	10 mg.kg	-
Bleomycin [28]	F_1_(C57BL/6JxCBA/Ca)	15–30 mg/kg	+
Cyclophosphamide [28]	F_1_(C57BL/6JxCBA/Ca)	40–80 mg/kg	+
Mitomycin C [28]	F_1_(C57BL/6JxCBA/Ca)	2.5–5 mg/kg	+
Procarbazine [28]	F_1_(C57BL/6JxCBA/Ca)	50–100 mg/kg	+
Etoposide [26]	CBA/Ca	80 mg/kg	+
**Chemical mutagens:**			
Ethylnitrosourea [26]	CBA/Ca	12.5–75 mg/kg	+
Isopropyl methanesulfonate [26]	CBA/Ca	12.5–37.5 mg/kg	+
Extremely low frequency magnetic fields [29]	F_1_(BALB/cxCBA/Ca)	10–300 μT	-

^1^ Significant mutation induction (+). No significant effects of exposure (−).

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
