# Peer review of "Mutation Induction in Humans and Mice: Where Are We Now?"

_cancers, 2019, doi:10.3390/cancers11111708_

Round 1

Reviewer 1 Report

The author has adequately addressed all my major comments and corrected the minor errors.

Three more minor points:

The sentence added in line 49 is redundant to that in line 50. Keeping only one of them will be fine. Line 17, "than in the offspring" should it be "that in the offspring". I'm still not sure why the "Conclusions" section is numbered as subsection 5, directly following subsection 2.2. Please check. 

Author Response

I ma grateful the the reviewer for spotting these typos. They have been corrected.

Reviewer 2 Report

The author has adequately addressed my comments in the revised version.

Author Response

Thanks for your positive feedback.

This manuscript is a resubmission of an earlier submission. The following is a list of the peer review reports and author responses from that submission.

Round 1

Reviewer 1 Report

This is an excellent review providing an overview of the current state of the art of mutation induction in humans. It is interesting, well written and clearly described.

The author may consider adding some figures/schematics to the review to illustrate the methods described as these may not be familiar to all readers (myself included) – tandem repeat human minisatellite loci/minisatellite mutation rate and expanded tandem repeat loci/ ESTRs, genome wide sequencing.

On page 2, after the sentence on line 52, please add some explanation or comment on the conflicting results.

On page 2, consider changing the title of section 2.2 to Transgenerational instability

Minor edits

Page 1, line 36, change ‘in odds’ to ‘at odds’

Page 1, line 42, change ‘With this respect’ to ‘In this respect’

Page 1, line 44, change ‘..at these loci by several orders of magnitude exceeds those for protein coding genes’ to ‘..at these loci exceeds those for protein coding genes by several orders of magnitude’

Page 2, line 49, ‘a significant increase was detected’ - compared to what?

Page 2, line 69, change ‘..exposure to them..’ to ‘…exposure to these drugs’

Page 3, line 94, change ‘..close to than..’ to ‘..close to that..’

Page 3, line 97-99, change ‘The results of recent study….in spectrum of…to chemical mutagen…’ to ’ ‘The results of a recent study….in the spectrum of…to the chemical mutagen…

Page 3, line 110, change ‘..relative..’ to ‘..relatively..’

Page 4, line 134, change ‘..summarised..’ to ‘..summarises..’

Page 4, line 154, change ‘..wide range..’ to ‘..a wide range..’

Page 6, line 206-207, change ‘…aimed to establish the mecahnisms….and well as…are clearly warranted’ to ‘…aimed at establishing the mecahnisms….as well as… is clearly warranted’

Author Response

I am very much grateful for all the suggestion made by the Reviewer. All typographic errors have been corrected.

Reviewer 2 Report

This is a very nice review summarizing decades of research in mutation induction in humans and mice, from both the author’s own group and many other researchers. The author first described the effects of various mutagens on genomic regions with particularly high mutation rates, such as minisatellites and expanded simple tandem repeat loci (ESTR), from both human and mouse. He then extended the discussion to more recent findings regarding the genome-wide mutagenic effects of ionizing radiation enabled by whole-genome sequencing. In the next section, the author discussed the intriguing finding in mice of transgenerational genome instability in non-exposed offspring of males exposed to radiation or mutagenic reagents. The author concluded with potential of whole-genome sequencing approach in future study of genetic risks of exposure to ionizing radiation and other mutagens, highlighting the need to further explore whether transgenerational genome instability exists in humans and the potential mechanisms.

Overall, I find this review timely and of considerable interest, as research in germline mutagenesis has made tremendous progresses, thanks to recent advances in sequencing technologies and computational methods. With basic knowledge such as the baseline frequency and spectrum of germline mutations at hand, it is natural and important to ask what factors influence the germline mutation rate and could lead to inter-individual variation in humans. The overall structure of this review is clear and easy to follow, and the text well written. I have a few questions and suggestions in specific parts of the paper that I hope the author could consider and further clarify. In addition, I wish the author could discuss the conflicting results of some studies and incorporate findings from various different experiments to provide a more integrated overview of this field of research.

Major points:

Similarities and differences in germline mutation patterns and mechanisms in humans and mice.

According to the title, this review is focused on mutation in humans, but at least half of the experimental results come from mice. Therefore, I would like to see a brief discussion of (1) why a mouse model is commonly adopted; (2) to what extent the mutation properties and mechanisms are conserved between human and mouse (is there any surprising differences?); (3) potential ways to validate findings from mice in humans. Alternatively, the author can modify the paper title to be include both human and mouse, or focus on mammals.

The results of a subset of studies are not clearly stated.

Sometimes experiments are described without the results being clearly stated. For example, in lines 61-63, a series of experiments are mentioned regarding the effects of various mutagens on ESTR loci, but the reader is left wondering whether all these mutagens have significant effects and what are the magnitudes of the effects.

More generally, since various experiments are presented, some in humans and some in mice, some showing positive results and some negative results, I would really appreciate a systematic summary of all the experiments and findings in one table, including the organism studied (human or mouse), the mutagen used (ionizing radiation, chemical mutagen, anticancer drug, etc), the dosage, and the result (significant elevation in mutation rate, shift in mutation spectrum, etc).

Potential explanations for seemingly contradicting results.

In lines 45-54, both positive and negative results are found in studies of families that are exposed to ionizing radiation. Could the author provide interpretation of the results and potential explanations for these seemingly discrepancies?

Contradicting description of the experiment results.

I might have misunderstood something, but I find parts of the text directly contradicting each other: lines 155-157 say “the manifestation of transgenerational instability among the F1 offspring of male mice exposed to the abovementioned anticancer drugs does NOT substantially differ from than in offspring of irradiated males”, while Figure 3(a) clearly shows significant higher mutation rates from the control in all test groups. Even more confusing, the author later writes that “On the other hand, paternal exposure to clinically-relevant doses of anticancer drugs can significantly destabilize the F1 genomes (Fig 3a).”

So… is there significant effects or not? This seems to be a key result from the author’s own group. Why is there such inconsistency in the text (and the figure)?

Transgenerational mutagenic effect of paternal exposure versus maternal exposure

I find the observations described in section 2.2 fascinating, but it seems the experiments are limited to the configuration of exposed fathers plus unexposed mothers. What are the effects of the reciprocal configuration, i.e., exposed mothers and unexposed fathers? Are there any synergistic effects when both parents are exposed?

If there is no transgenerational genome instability from exposed mothers, the result should be presented, as this is a meaningful negative result that suggests asymmetric contribution of father and mother to genetic risk of their offspring. If such transgenerational effect from the maternal side is not tested, this should also be clearly stated, which can promote future studies.

Minor points:

In the abstract, the two sentences “Here I briefly summarize the results of recent studies on the pattern of mutation induction in the human and mouse germline. The results of recent studies on the genome-wide effects of exposure to mutagens on mutation induction in the mammalian germline are presented and discussed” are highly similar, and it is unclear why the author needs two sentences to express the same meaning. Only upon reading the text, I begin to realize that earlier studies mostly focused on minisatellites and ESTRs, and that only until recently have researchers started to look at the genome-wide effects of mutagens. I hope the author to differentiate these two sentences in the abstract to highlight the differences in time and methodology. Line 22, Page 1: it should be “one of the key risk factor­S” or “one key risk factor”. Line 36, Page 1: it should be “at odds” instead of “in odds”. Line 59, Page 2: it should be “expanded SIMPLE tandem repeat loci”, otherwise the acronym ESTR lacks a letter “S”. Line 94, Page 3: it should be “close to thaT in controls” instead of “close to than in controls”. Line 108, Page 3: a comma is needed after the parentheses. Line 126, Page 4: it will be better to present the magnitude of change and corresponding confidence intervals than just stating “highly significantly”. Line 156, page 5: it should be “differ from thaT” instead of “differ from than”. The section title jumps from “2.2 Is it all?” to “5. Conclusions”. What are sections 3 and 4? In addition, the title of section 2.2 is not very informative, could be author consider a more explicit title such as “Transgenerational genome instability induced by mutagens in mouse”?

Author Response

The title has been changed according to the suggestion made by the Reviewer.

An extra table summarizing the results of our ESTR studies has been included in the text.

A statement regarding the negative results obtained in studies on minisatellite mutation rates in the irradiated human families has been added.

As far as the Reviewer’s comment regarding the results shown on Fig 3a is concerned, I believe that there is some sort of misunderstanding. On this graph the ratio of ESTR mutation frequencies in the offspring of exposed males to that in control animals is shown. Judging from the results showing on this graph, in the F1 offspring of male mice exposed to ionizing radiation and anticancer drugs roughly similar, 2-fold and significant increases in ESTR mutation frequencies in all the F1 offspring of exposed males. An extra sentence further clarifying this issue has been added to the figure legend.

The Reviewer also raised the important issue of the transgenerational effects of maternal irradiation, which has been clarified in the Chapter 2.2.

All typographic errors have been corrected.